# The number of olfactory stimuli that humans can discriminate is still unknown

Richard C Gerkin[1]*, Jason B Castro[2,3]

[1]School of Life Sciences, Arizona State University, Tempe, United States; [2]Department of Psychology, Bates College, Lewiston, United States; [3]Program in Neuroscience, Bates College, Lewiston, United States

**Abstract** It was recently proposed (*Bushdid et al., 2014*) that humans can discriminate between *at least* a trillion olfactory stimuli. Here we show that this claim is the result of a fragile estimation framework capable of producing nearly any result from the reported data, including values tens of orders of magnitude larger or smaller than the one originally reported in (*Bushdid et al., 2014*). Additionally, the formula used to derive this estimate is well-known to provide an upper bound, not a lower bound as reported. That is to say, the actual claim supported by the calculation is in fact that humans can discriminate *at most* one trillion olfactory stimuli. We conclude that there is no evidence for the original claim.

*For correspondence: rgerkin@asu.edu

**Competing interests:** The authors declare that no competing interests exist.

**Reviewing editor**: Alexander Borst, Max Planck Institute of Neurobiology, Germany

## Introduction

A recent paper (*Bushdid et al., 2014*) proposed that humans can discriminate between at least a trillion olfactory stimuli. Using that paper's methods to reanalyze the data it presented, we show that this estimate is problematically fragile. Specifically, it varies systematically and sensitively (over tens of orders of magnitude, in both directions), for very modest changes in incidental experimental and analysis parameters against which a result ought to be robust. Had the experiment enlisted ~ 100 additional subjects similar to the original ones, the same analysis would have concluded that *all possible stimuli* are discriminable (i.e., that each of the more than $10^{29}$ olfactory stimuli possible in their framework are mutually discriminable). By contrast, if the same experimental data were analyzed using moderately more conservative statistical criteria, it would have concluded that there are fewer than 5000 discriminable olfactory stimuli—no larger than the folk wisdom value that the new estimate purports to replace.

Therefore, under this framework, data describing the same underlying perceptual abilities admit a wide range of extremely disparate (varying over *25 orders of magnitude*), yet unobjectionable alternative conclusions (including both the largest and smallest possible estimates allowed by the analysis framework). We conclude that the framework is unsound: there may be trillions of discriminable olfactory stimuli, or more, or fewer, but the framework does not provide the means for settling this question. Here we first demonstrate the framework's fragility, and then explain the sources of that fragility. For most of this paper, we remain agnostic about whether the framework is conceptually sound, to highlight the fact that it has strictly methodological problems of a statistical origin that do not depend on the validity of a competing set of assumptions.

We also show that the formula used to derive the estimated number of discriminable stimuli, given an estimated perceptual limen, yields an upper bound, not a lower bound, meaning that any estimate derived here or in (*Bushdid et al., 2014*), under any assumptions, is a maximum and not a minimum. In other words, the original paper in fact supports the conclusion that humans can discriminate *at most* one trillion olfactory stimuli (or more or fewer, due to the problem described above), a rather uninspiring claim. In a concluding section, we explore possibilities for improving the estimate.

**eLife digest** Scientists are interested in the number of colors, sounds and smells we can distinguish because this information can shed light onto how our brains process these senses both in health and disease. It is relatively straightforward to determine how many colors we can see or sounds we can hear because these stimuli are well defined by physical properties such as wavelength. We know the range of wavelengths that the eye can see or the ear can hear, and we can also understand how two such stimuli (e.g., red and blue) are arranged perceptually (think of a color wheel). It is harder, however, to do the same for smell because most 'olfactory stimuli' consist of mixtures of different odor molecules. Moreover, we understand much less about how olfactory stimuli are arranged perceptually.

In 2014 researchers at Rockefeller University reported that humans can distinguish more than one trillion smells from one another. To calculate this number the researchers tested the ability of human subjects to discriminate between mixtures of different odor molecules. Each mixture consisted of 10, 20 or 30 molecules selected from a chemical library of 128 different odor molecules. Since each mixture of 10 molecules could contain any 10 of the 128 molecules, more than 200 trillion combinations were possible; the number of possible combinations for the 20- and 30-molecule mixtures were even higher.

The aim of the experiment was to identify—by sampling from this very large number of combinations—the number of molecules that two mixtures could have in common and still be distinguishable to the typical person. The Rockefeller team used this number and a geometrical analogy to conclude that humans could discriminate at least 1.72 trillion odors, which was much higher than expected from previous reports and anecdotes.

Now Gerkin and Castro report that the claims made in the Rockefeller study are unsupported because of flaws in the design of the analytical framework used to make sense of the data. In particular, Gerkin and Castro report that the results are extremely sensitive to some parameters of the experimental and analytical design, such as the number of subjects tested, whereas the results of a robust analysis would not be so sensitive to such factors. By modestly varying any of these parameters it is possible to obtain almost any value for the number of smells that can be discriminated. Moreover, the geometrical analogy used set an upper bound on the final answer, rather than a lower bound: in other words, even assuming that the rest of the analysis was robust, the result should have been that humans can discriminate 'no more than' 1.72 trillion smells rather than 'at least'. In a separate paper Meister also reports that the 1.72 trillion smells claim is unjustified.

## Problems with the estimate

The first main concern is that the estimated number of discriminable stimuli depends steeply, systematically, and non-asymptotically on choices of arbitrary experimental parameters, among them the number of subjects enrolled, the number of discrimination tests performed, and the threshold for statistical significance. We show below that the order of magnitude claim of 'one trillion olfactory stimuli' requires that those parameters assume a very narrow set of values. Certainly, the precise value of an estimate may change as additional data are collected, but the estimate should not change *in expectation*; it should not be possible to make an estimate arbitrarily large (or small), simply by collecting more (or less) data. Similarly, the estimate itself should not become arbitrarily small or large with adjustment of a significance criterion. Estimates that scale systematically with such incidental parameter choices are considered statistically *inconsistent* (*Figure 1*). It is the inconsistency of the present estimate that produces a tremendously large space of extremely different, yet unobjectionable alternative conclusions that can be reached about the number of discriminable olfactory stimuli.

To illustrate that we can correctly recapitulate the analysis undertaken in (*Bushdid et al., 2014*), *Figure 2* shows our reproduction (using raw supplementary data from [*Bushdid et al., 2014*]) of two critical figures from that paper (*Bushdid et al., 2014*), from which its main conclusion was drawn. See *Table 1* for definitions of parameters used here and in (*Bushdid et al., 2014*). *Figure 3* and *Table 2* quantify the fragility of this conclusion, by generating estimates using the same framework under trivial alternative scenarios in which different numbers of subjects (or mixtures) were used, or different choices of statistical threshold ($\alpha$) were used for assessing discriminability. Thus, we produced all

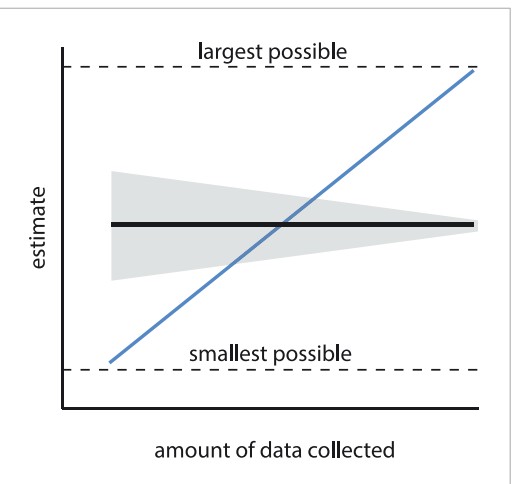

**Figure 1**. Consistency of an estimator. An estimator is consistent if the resulting estimate asymptotically converges (in expectation) as sample size increases (black line). Uncertainty in the estimate (gray area) may shrink with sample size, but the estimate itself should not systematically change with sample size, and should converge on the truth. Estimators without this property are termed inconsistent (the blue line is a relevant example), and are considered unreliable, as the resulting estimate can be heavily biased by the sample size. If the estimate has a minimum and maximum allowed value (see *Equation 1*), an especially inconsistent estimator can even produce any estimate within that range.

The following figure supplement is available for figure 1:

**Figure supplement 1**. Fraction discriminated at which statistical significance is reached.

values shown here by analyzing the data from (*Bushdid et al., 2014*), using the methods described therein, and varying only parameters. Code to reproduce these and all subsequent analyses is available at http://github.com/rgerkin/trillion, documented at http://nbviewer.ipython.org/github/rgerkin/trillion/blob/master/journal.ipynb.

In *Bushdid et al., 2014*'s experimental framework, there are three sets of experiments, varying in the number of distinct molecular components $N$ per mixture tested. We consider the $N = 30$ case (without loss of generality) for which there are $\sim 10^{29}$ possible olfactory stimuli, and for which the smallest possible number of discriminable stimuli is $\sim 4500$ (see *Equation 1* below). *Figure 3* and *Table 2* thus demonstrate that (1) there is a regime of reasonable parameter choices for which one concludes that all possible olfactory stimuli (i.e., all $\sim 10^{29}$ of them) are discriminable; and (2) there is another regime of reasonable parameter choices for which one concludes that the smallest possible number of stimuli (i.e., only $\sim 4500$) are discriminable. The only assumption required to obtain these estimates is that performance in new subjects is similar to performance in the original subjects.

The fragility of the conclusion results from the claim in (*Bushdid et al., 2014*) that a modest (if very interesting) correlation—between the discriminability of a pair of mixtures and the overlap (fraction of shared components) of those mixtures—is evidence that a *particular degree* of mixture overlap defines a boundary that partitions the discriminable from the indiscriminable in a very high-dimensional space. Below, we explore the consequences of this decision, and its implications for calculating the number of discriminable olfactory stimuli.

## Explanation of the problems with the estimate

### Recap of the basic framework

The framework's logic is built on an analogy to color vision, where estimating the number of discriminable colors requires knowing only two numbers: the size of the stimulus space (that is, the range of visible wavelengths), and the minimally discriminable distance between a typical pair of stimuli (*Figure 4*). Dividing the first number by the second amounts to asking how many discriminable intervals can be 'packed' into the stimulus space, with that number providing an estimate of the number of discriminable color stimuli.

Because olfactory stimuli do not have obvious physical dimensions analogous to wavelength, olfaction is not amenable to an identical calculation. Instead, (*Bushdid et al., 2014*) established a theoretical framework that yielded a similar calculation based upon the same underlying idea. (*Bushdid et al., 2014*) proposed to divide the size of a investigator-determined olfactory stimulus space by a data-determined variable representing resolution in this space. Instead of being continuous, one dimensional, and defined by some intrinsic stimulus variable like wavelength, the olfactory stimulus space was defined to be the discrete, high-dimensional space spanned by all mixtures containing $N = 30$ different components (molecules) that could be assembled from a library

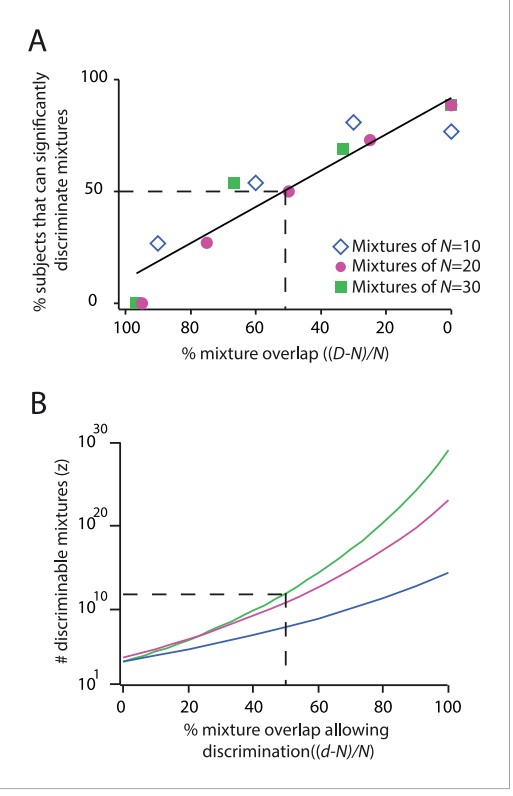

**Figure 2**. Reproduction of the main result published in (**Bushdid et al., 2014**), from analysis of raw data made available in supplemental materials of (**Bushdid et al., 2014**). Compare to **Figures 3, 4** in that publication. (**A**): Discriminability vs mixture overlap, expressed as a percentage of the mixture size N. From this analysis, (**Bushdid et al., 2014**) derives $\frac{d-N}{N} \sim 51\%$ (vertical dashed line) as the critical value of mixture overlap at which 50% of mixtures achieve 'significant discriminability'. (**B**): Estimated number of discriminable mixtures z vs mixture overlap (expressed as a percentage of N) allowing discrimination. The plot is obtained by regression and interpolation of results in **A** combined with **Equation 1**, with colors corresponding to values of N as shown in **A**. For a value of $\sim 51\%$ as derived in **A**, one obtains the 'trillions' figure reported in (**Bushdid et al., 2014**).

The following figure supplement is available for figure 2:

**Figure supplement 1**. Reconstruction of percent correctly discriminated using raw data from (**Bushdid et al., 2014**).

of C = 128 molecules; (**Bushdid et al., 2014**) also considers the N = 10 and N = 20 cases, which we ignore in this section with no loss of generality. This space of possible mixture stimuli is astronomically large $\binom{C}{N}$, owing to the proverbial 'combinatorial explosion', and each point in the space corresponds to a specific multi-component mixture.

One definition of distance between stimuli in this space is the number of components D by which the stimuli differ. For example, nearest neighbors would be stimuli sharing all components but one (D = 1), and the most distant points in this space would be stimuli differing in all components (D = N).

(**Bushdid et al., 2014**) showed that discriminability of a stimulus pair tends to increase with the distance D between the stimuli in that pair (**Figure 2A**), and then argued for the existence of a special distance d corresponding to the D at which stimuli are 'just discriminable'. In other words, for D > d stimuli should more often than not be considered discriminable and for D < d they should more often than not be considered indiscriminable. By calculating d, one could in turn readily calculate the number of stimuli within a distance D ≤ d of a typical point in the stimulus space using the provided formulas. Geometrically, the set of stimuli with distance D ≤ d from a reference stimulus corresponds to a filled 'ball' of stimuli indiscriminable from the reference stimulus at its center. Conversely, the reference stimulus should be discriminable from stimuli outside the ball. We could thus count the number z of non-overlapping balls that can be packed into the stimulus space, as proposed in (**Bushdid et al., 2014**), by analogy to the example for color vision:

$$z(d) = \frac{\binom{C}{N}}{ball(d/2)} \tag{1}$$

where 'ball' is defined as:

$$ball(r) = \sum_{x=0}^{r} \binom{N}{x}\binom{C-N}{x} \tag{2}$$

**Equation 1** produces the final estimate z of the number of discriminable stimuli. Note that while this has been interpreted as 'the answer' to the sphere packing problem in high dimensions, it is in fact only a best-case scenario (an upper bound). The *exact* number of d-spanning spheres that can be packed in a discrete space defined by a particular C and N has in fact only been computed for a few specific, modest cases of these values. In general, it is only possible to report bounds for these values. This is discussed at more length in the section. 'An upper or a lower bound?', below, as well in the supplemental materials.

**Table 1**. Definitions of parameters

| | |
|---|---|
| $z$ | Estimated number of discriminable olfactory stimuli |
| $C$ | Number of distinct compounds available to make mixtures |
| $N$ | Number of distinct compounds in a mixture |
| $O$ | Number of distinct compounds shared by a mixture pair |
| $D$ | Number of distinct compounds in one mixture of a pair that are not shared by the other. ($D = N - O$) |
| class | All mixture pairs with the same value of $N$ and $D$. |
| $d$ | The value of $D$ for which mixture pairs of a given $N$ are more likely than not to be discriminable at a rate significantly above chance. |

$C$ and $N$ are fixed by experimenter choices, and $d$—the resolution-like term—is the only quantity derived from data that is related to measured psychophysical performance. Note that for $C = 128$, $N = 30$, as used in (*Bushdid et al., 2014*), the *largest* and *smallest* possible values this equation can produce are $\sim 1.5 \times 10^{29}$ (for $d = 0$) and $\sim 4500$ (for $d = N$), respectively. Assuming this framework is conceptually unproblematic (but see *Meister, 2015*), the only question becomes: How do we derive $d$ from the data?

## Derivation of the critical parameter *d*

### Thresholding the fraction discriminated

A classic psychometric curve (*Figure 4B*), showing discriminability as a function of inter-stimulus distance $D$, admits a few plausible ways to derive $d$. The simplest is to use a discriminability threshold, such that $d$ corresponds to the distance $D$ at which the 'fraction correct' reaches a certain value. In (*Bushdid et al., 2014*)'s three-alternative forced-choice experiments, chance responding would produce a fraction correct of $\frac{1}{3}$, so the appropriate threshold would be somewhere between $\frac{1}{3}$ and 1. This threshold choice would be arbitrary—we might say that a fraction correct of $\frac{1}{2}$ reflects discriminability, or alternatively we might choose $\frac{2}{3}$ or any other value between $\frac{1}{3}$ and 1.

If the psychometric curve is sufficiently steep near some value of $D$ (*Figure 4—figure supplement 1A* represents an ideal case) then the derived $d$ will vary minimally over a wide range of choices for the threshold. In this scenario, we might be confident that the $d$ we derive is a truly meaningful measure of resolution—it would be robust. If not (*Figure 4—figure supplement 1C*), it will be very fragile. We explored this approach (*Figure 4—figure supplement 2*), and concluded that it does not suffice for deriving a robust $d$.

### Thresholding the fraction *significantly* discriminable

The approach actually used in (*Bushdid et al., 2014*) is instead to apply a threshold not to the fraction *discriminated* (explored in *Figure 4—figure supplement 2*), but to the fraction *significantly discriminable*. In other words, determine for which subjects (or alternatively, for which classes of mixtures) the fraction discriminated is *significantly greater* than $\frac{1}{3}$, i.e., for which subjects the null hypothesis of chance discrimination can be rejected. To facilitate visualization of this step, (*Bushdid et al., 2014*) re-plotted the summary data (fraction correctly discriminated) as fraction significantly discriminable (*Figure 2A*). This view of the data provides a linear relationship between distance $D$ and the fraction significantly discriminable, which holds across all the values of $N$ tested. The relationship is much steeper than for fraction discriminable (compare *Figure 2* and *Figure 4—figure supplement 2*) because this hypothesis-testing step acts as a strong non-linear threshold that exaggerates otherwise small differences in the data. An arbitrary choice of threshold is required; (*Bushdid et al., 2014*) chose a threshold of 50% significantly discriminable, and computed $d$ from the fraction significantly discriminable using linear regression and interpolation.

Varying the threshold (i.e., 50%) itself (not shown), would change the computed $d$ (and consequently $z$), but this is not the largest issue. By introducing a hypothesis-testing step, the $d$ derived from *Figure 2* now varies systematically with the number of subjects enrolled in the study (and the number of mixtures tested), and with the choice of significance criterion $\alpha$. This is because each data point used to compute $d$ becomes the binary result of a hypothesis test, each

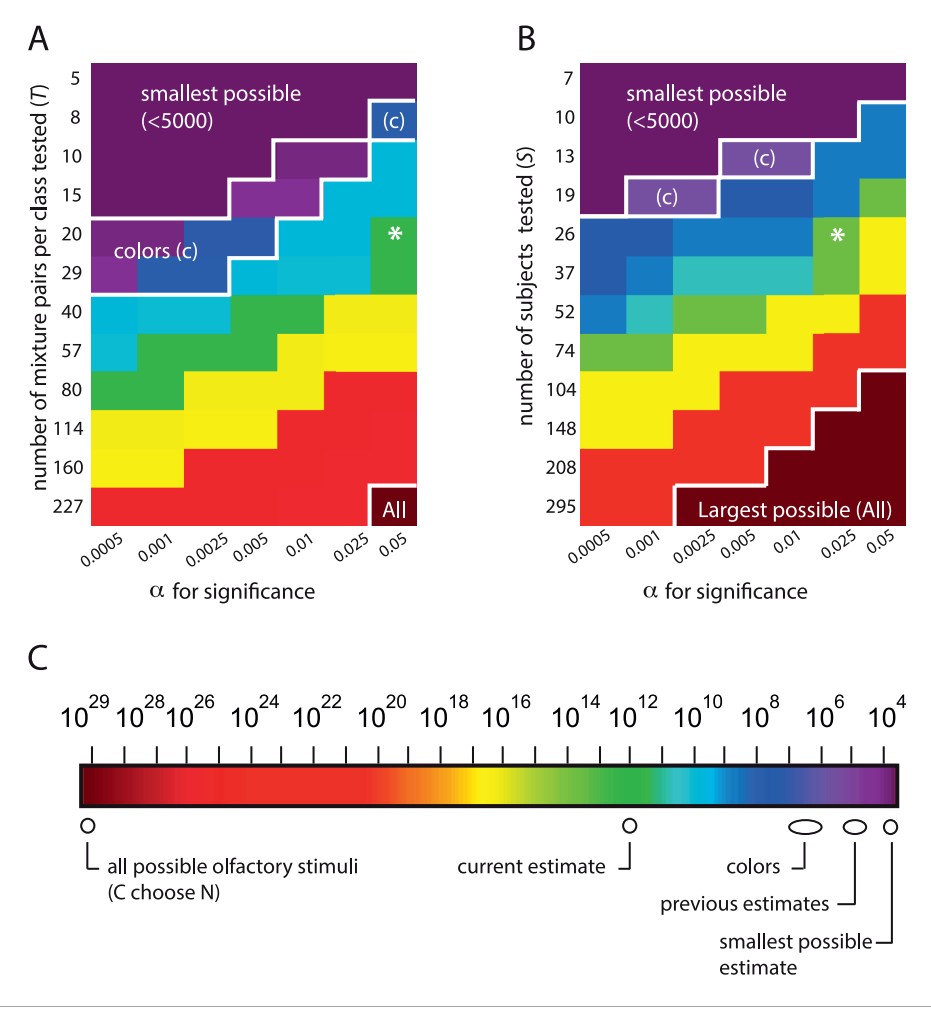

**Figure 3**. The estimation framework supports nearly any alternative conclusion, including the smallest and largest estimates possible under the framework. (**A**): Heat map showing alternative conclusions reached for different choices of $T$, the number of mixture pairs per class to test, and application of alternative significance threshold $\alpha$ for discriminability, with the data from (**Bushdid et al., 2014**). Asterisks (*) show the parameter regime ($T = 20$ mixtures, $\alpha = 0.05$) used in (**Bushdid et al., 2014**). Other values on each axis are chosen in a geometric progression around those parameters. The contour in the lower right labeled 'All' demarcates a regime in which one will conclude that the largest possible number of mixture stimuli (i.e., all $z(d=0) = \binom{128}{30} > 10^{29}$ of them) are discriminable (see **Equation 1**). The contour in the upper left labeled 'smallest possible' demarcates a regime in which one will conclude that the smallest possible number of stimuli are discriminable, that is, only $z(d = N = 30) < 5000$ of them. The contour labeled 'colors' demarcates a regime in which one concludes that the number of discriminable olfactory stimuli is the same order of magnitude as the number of discriminable colors. (**B**): Heat map similar to left, only with number of subjects on the vertical axis. A choice of $\alpha = 0.025$ is necessary to obtain the estimate that (**Bushdid et al., 2014**) reports for this analysis. (**C**): Colorscale for **A** and **B**, with reference landmarks.
The following figure supplement is available for figure 3:

**Figure supplement 1**. Steep, systematic, and non-asymptotic dependence of the estimate on sample size ($S$ or $T$) and threshold $\alpha$ for statistical significance.

of which depends critically on sample size and test specificity. Because $d$ is then fed into an expression (**Equation 1**) that explodes geometrically, the result is a recipe for producing any of a range of estimates for $z$ that one might choose. If one enlists more subjects or slackens the

**Table 2.** Estimates of $z$, the number of discriminable olfactory stimuli, for different possible parameters values, for the $C = 128$, $N = 30$ case used in (**Bushdid et al., 2014**)

**A**

| # Discriminable stimuli ($z$) | Significance threshold ($\alpha$) | # Tests per class ($T$) |
| --- | --- | --- |
| $2.02 \times 10^{12}$ | 0.05* | 20* |
| $4.56 \times 10^3$† | 0.05* | 5 |
| $1.54 \times 10^{29}$‡ | 0.05* | 185 |
| $8.94 \times 10^3$ | 0.001 | 20* |
| $1.79 \times 10^4$ | 0.01 | 15 |

**B**

| # Discriminable stimuli ($z$) | Significance threshold ($\alpha$) | # Subjects ($S$) |
| --- | --- | --- |
| $3.81 \times 10^{13}$ | 0.025* | 26* |
| $4.56 \times 10^3$† | 0.025* | 7 |
| $1.54 \times 10^{29}$‡ | 0.025* | 135 |
| $3.47 \times 10^7$ | 0.001 | 26* |
| $2.98 \times 10^5$ | 0.01 | 15 |

This recapitulates selected points from **Figure 3**.
* Indicates that the parameter value was used in (**Bushdid et al., 2014**). We assume here that new subjects perform similarly to the original subjects.
Note that $4.56 \times 10^3$ (†) and $1.54 \times 10^{29}$ (‡) are the smallest and largest possible values allowed by the framework from (**Bushdid et al., 2014**).

significance criterion, a very large (even the largest possible) number will be obtained. If one enlists fewer subjects or makes the significance criterion more strict, a very small (even the smallest possible) number will be obtained. **Figure 3—figure supplement 1** shows the explicit dependence of the estimate on each of these quantities alone. Naturally, these can be varied in tandem too, with even more dramatic consequences, as described above (**Figure 3** and **Table 2**).

A hypothesis test is meant to assess the strength of evidence for or against a hypothesis (often against a null hypothesis), not to make a point estimate. However, it may not be uncommon for researchers to use hypothesis testing in the manner done in (**Bushdid et al., 2014**)—to count the number or fraction of data points exhibiting a certain property. In many cases this may amount to a venial statistical sin with (hopefully) benign consequences. But that is unfortunately not the case in (**Bushdid et al., 2014**), due in part to the extremely steep dependence of $z$ on $d$ guaranteed by **Equation 1**.

If one claims that an estimate is meaningful, it is fair to ask how vigorously would one have to defend a specific choice of arbitrary experimental parameters to defend a particular order-of-magnitude range around that estimate. Unfortunately, the systematic sensitivities exhibited here severely undermine the plausibility and relevance of the estimate reported in (**Bushdid et al., 2014**). Due to these sensitivities, one could pick almost any number of discriminable stimuli in advance, and affirm this number using these or similar data. Ultimately, the absence of a robust $d$ to characterize the data is an insurmountable obstacle for the framework.

## Building the stimulus space

### The structure of the stimulus space
One might ask: what is the right way to calculate $d$ in order to obtain a robust estimate of the number of discriminable stimuli? Before heading down this road and devising alternative statistical approaches, it is worth first clearly articulating the assumptions of a framework in which a single variable plays such a special role. Under what conditions is it sensible to expect that plugging a single

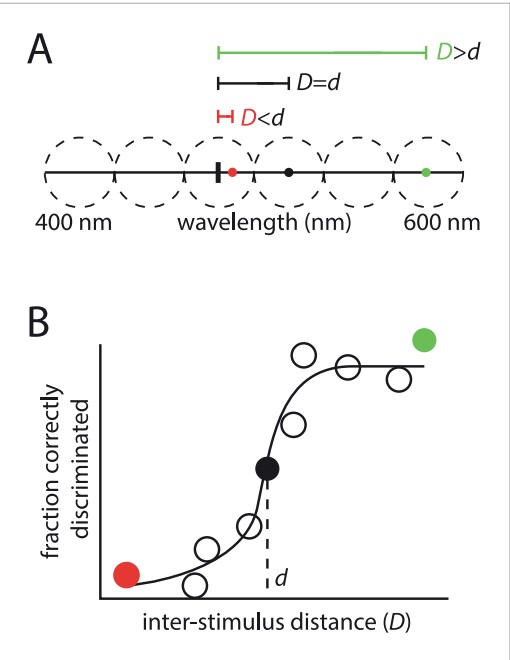

**Figure 4**. 'Sphere packing' to estimate the number of discriminable colors: the motivation behind the framework in (*Bushdid et al., 2014*). (**A**): Hypothetical example showing a range of visible wavelengths. Relative to a reference stimulus (thick vertical tick mark), extremely distant stimuli (green circle) in this space are easy to discriminate, whereas extremely close stimuli (red circle) may be impossible to discriminate, as they are beyond the resolution of color vision. At some critical inter-stimulus distance, $d$, stimuli will be 'just discriminable' (black circle). A typical stimulus pair on the space, separated by distance $D$, will tend to be discriminable if $D > d$, and indiscriminable if $D < d$. (**B**): This partitioning into discriminable and indiscriminable sets is captured in the sigmoidal shape of the psychometric curve plotting discriminability vs distance. Knowing that an interval of length $d$ on the space will tend to span 'just discriminable' stimuli, one can calculate how many such intervals, $z$, can be 'packed' onto the space to estimate the number of discriminable colors.

The following figure supplements are available for figure 4:

**Figure supplement 1**. Behavior of psychometric curves for hypothetical data describing discriminability vs inter-stimulus distance.

**Figure supplement 2**. Can the fraction discriminated be used to measure $d$ directly, without resorting to hypothesis testing?

data-derived number ($d$) into *Equation 1* will produce a meaningful estimate of the number of discriminable olfactory stimuli?

To gain some intuition into this, we can ask the analogous question in the simplified visual system example (*Figure 4*) that was used as the principal motivation for the procedure. The 'sphere packing' calculation in this case naturally involves measuring the resolution of perception in terms of the stimulus, but its validity is not a consequence of this measurement alone. Rather, the procedure in *Figure 4* is sensible because the thing we are calling an independent stimulus dimension (wavelength) is respected as such by perception: we encounter monotonically changing, non-redundant percepts as we move from one extreme of the stimulus space to the other. If we didn't—say, if the same percept 'blue' were experienced for several non-overlapping disjoint intervals—the sphere packing formulation would fall apart. We might observe that on average discriminability improves with distance, but this would not be evidence of a characteristic length scale that partitions stimulus pairs into discriminable vs indiscriminable sets.

Thus the sphere-packing framework is valid only if the underlying geometry of *stimulus space* (that the investigator has designed) aligns with the geometry of *perceptual space* (as implemented in neural circuitry). Formally, the map from stimulus space to perceptual space needs to be homeomorphic, or nearly so. See (*Meister, 2015*) for further insight on this issue.

## Redundancy in the stimulus space

Instead of providing evidence for this homeomorphism, it was assumed in (*Bushdid et al., 2014*) for the purposes of calculation that each component of the molecular library (of size $C = 128$ in [*Bushdid et al., 2014*]) spanned an informative additional dimension for perception to explore: each molecule in the library is treated as an olfactory primary that is independent of all the others. This is the assumption, codified in the numerator of *Equation 1*, that allows for a massive space of potential discriminable stimuli. Indeed, the guaranteed runaway growth of the numerator as molecules are added to the $C$-sized library was offered in (*Bushdid et al., 2014*) as an argument for why the reported 'trillion' figure is an underestimate—after all, $C$ could always be higher.

It is worthwhile to quantify the behavior of the estimate as $C$ changes. First, the estimate depends geometrically on $C$, with a power law exponent of $\sim 30$ (*Figure 5*, blue line). In other words, if the chemical library were doubled, the estimate $z$ would increase by a factor of $2^{30}$ under constant

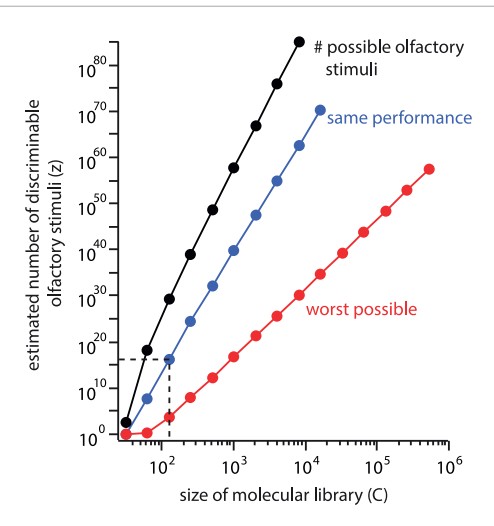

**Figure 5**. Explosive growth of the estimate $z$ on the size ($C$) of the molecular library. The number of possible stimuli $z$ that can be assembled by choosing $N = 30$ distinct molecules from a library of size $C$ increases geometrically with $C$ (black line). If a library of a different size had been used, and similar subject performance resulted, the estimated number of discriminable stimuli $z$ would grow along a similar trajectory (blue line). Even if performance deteriorated as $C$ increased, the estimate could never fall below the red line, which represents worst-case performance ($d = N$). This results from the combinatorial explosion inherent in *Equation 1*.

performance. If the component library were increased to the size of a standard flavor and fragrance catalog ($\sim 2000$ chemicals), the estimate would increase to $z \sim 10^{41}$, implying a unique olfactory percept for each carbon atom on earth.

Subjects' performance could become worse when mixtures are drawn from this larger, more complete library, and we acknowledge that we cannot know in advance what the newly calculated resolution $d$ would be on the new stimulus space. In other words, as the numerator of *Equation 1* increased, its denominator (given by *Equation 2*) might conveniently grow proportionally. Let us therefore assume that with a library of sufficient size, so many mixtures become indiscriminable that the resolution becomes as poor as the framework allows, with $d = N$. Even in this edge case, if only mixtures differing in all components were 'just discriminable', we would still calculate $10^{21}$ discriminable stimuli. If $C$ is increased to $10^6$, the smallest possible number of discriminable percepts (under the assumption of worst measurable performance, as above) is $10^{61}$, or 10 million trillion unique olfactory percepts for every carbon atom on earth (*Figure 5*, red line). One may object that the inflation of $C$ here is an unfair critique, as the perceptual redundancy of molecules must at some point provide an important constraint on the size of the artificially constructed stimulus space. Indeed, it has been reported that as few as thirty components are required to imbue most mixtures with a common smell, even when there is no component overlap between the mixtures (*Weiss et al., 2012*). But this is the essence of the problem with *Equation 1*: where does that point lie, and why wasn't the constraint important to consider for the original $C = 128$ molecular library?

## An upper or a lower bound?

Even if one takes the estimate of $d$ to be unimpeachable, the formula used to derive $z$ does not provide a lower bound as reported in (*Bushdid et al., 2014*). This much is suggested by the worst-case behavior of *Equation 1* as $C$ grows. After all, worst case behavior should correspond to $z = 1$. If one cannot discriminate anything (maximal $d$), then there is only one percept. Examining *Equation 1* more closely, we see that it is a variant of the so-called Hamming bound for constant weight codes (*MacWilliams and Sloane, 1977*). which is well-known to be an upper bound for an identically formulated problem in the theory of error-correcting codes. It is, as suggested in (*Bushdid et al., 2014*), an estimate derived from a hypothetical sphere-packing approach to filling the stimulus space, but it is the *largest* possible value for the correct answer, not the *smallest*. Hence, according to the Hamming bound, for $d = N = 30$ the upper bound on the number of discriminable stimuli is 4561, and we know the correct answer to be 1 (or 4, depending on conventions, see the Supplemental Materials). Since the upper bound exceeds the correct answer, *Equation 1*, while not particularly tight as an upper bound, is nonetheless not wrong, so long as we acknowledge that it is an upper and not a lower bound. The same applies for all other values of $d$, including the one derived from the data in (*Bushdid et al., 2014*).

Thus *Equation 1*, as used in (*Bushdid et al., 2014*), provides no insight into the *lower* bound for $z$, with a *lower* bound being required to overturn conventional wisdom about the number of discriminable stimuli. Instead, to obtain a lower bound one must dispense with the factor of 2 in

*Equation 1*, yielding Levenshtein's constant weight version of the so-called Gilbert-Varshamov bound for error-correcting codes ([*Levenshtein, 1971*; *MacWilliams and Sloane, 1977*; *Jiang and Vardy, 2004*],see Supplemental Materials). A plot of the lower bound obtained in this manner is shown in *Figure 6B*, along with the reconstructed upper bounds from (*Bushdid et al., 2014*) a, showing the true bounded interval for $z$. Intuitively, this corrected lower bound reaches $z = 1$ for worst-case $d$, implying sensibly that anosmics cannot discriminate any stimuli. In contrast, the upper bound (reported as a lower bound in 1) is on the order of several thousand for worst case $d$, showing that it cannot be a lower bound $d$; this can also be confirmed in *Figure 4* of (*Bushdid et al., 2014*).

## Avenues for improving the estimate

If one is seeking a conservative estimate of the number of discriminable stimuli in a perceptual space whose organization and intrinsic dimensionality are poorly understood, it is arguably more appropriate to use a model that accounts for the data with the smallest number of dimensions. The massive estimates possible in the framework of (*Bushdid et al., 2014*) are an immediate consequence of a definition of dimensionality driven by experimenter designation, not data.

We therefore propose an alternative framework: use experimental data to create a working map of the perceptual space, and then apply the sphere-packing framework to that map, rather than to a map of the stimulus space. In cognitive science, psychometrics, and marketing, subject responses to stimuli are used to create maps of the underlying perceptual (or conceptual) representations of those stimuli. These maps are characterized by the attribute that pairs of items which are considered intuitively to be perceptually near (rated similar or difficult to discriminate) are nearer to one another on the map than pairs of items which are perceptually more distant (rated dissimilar or easy to discriminate). There are many algorithms for generating such maps, many of which have been used before in olfaction, including variants of PCA (*Zarzo and Stanton, 2006*; *Khan et al., 2007*; *Koulakov et al., 2011*), non-negative matrix factorization (NMF, [*Castro et al., 2013*]), and multi-dimensional scaling (*Mamlouk et al., 2003*). While there are open questions in the generation of these maps (e.g., how many dimensions should they have?), they all have the virtue that their accuracy can be checked (e.g., by examining the correlation between subjects' indications of item pair dissimilarity and the distance

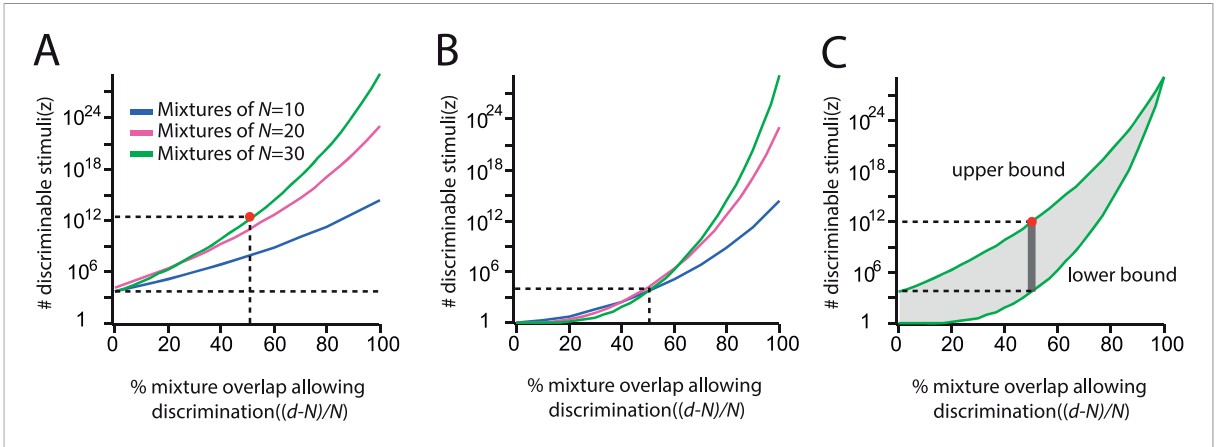

**Figure 6**. Upper and lower bounds of the number of discriminable stimuli. (**A**): Number of discriminable olfactory stimuli as a function of the estimated difference limen (the fractional mixture overlap allowing discrimination). This is simply the behavior of *Equation 1* as a function of $d$, for the three values of $N$ used in (*Bushdid et al., 2014*); the red dot (in both **A** and **C**) corresponds to the value reported in (*Bushdid et al., 2014*). The smallest possible estimate (thousands of stimuli) is indicated by the dotted line running the length of the abscissa (note also the y-intercept). As described in the text and in the supplement, this graph in fact shows the behavior of the upper bound (the so-called Hamming bound) for the mathematical problem of sphere packing. Compare with *Figure 3D* in (*Bushdid et al., 2014*). (**B**): Same plot as in **A**, only using the lower-bound for the same calculation. (**C**): Upper and lower bounds of the sphere packing problem for the N = 30case (green lines from **A** and **B**, respectively. The dark gray bar shows the range of defensible estimates under the sphere-packing framework, using the $d$ calculated in (*Bushdid et al., 2014*). Using that $d$, the number of discriminable stimuli may be as small as ~10,000, and is guaranteed to be no larger than ~1 trillion. Since the estimate of $d$ is also fragile (*Figure 3*), the data may in fact support any value in the shaded gray area.

between that pair on the map), and thus the maps can be improved. Developing these maps may also have the collateral benefit of revealing stimulus dimensions intrinsic to olfaction (if any), which could constrain the experimental choice of a resolution to measure.

Unfortunately, it is difficult if not impossible to create these maps from the data discussed here, because each mixture of a tested pair is used only once in (*Bushdid et al., 2014*), in that pair alone, and never in any other pairs. Thus, there are no serial comparisons of the same mixture that could be used to anchor a stimulus on the map relative to a stimulus against which it was not directly compared experimentally. Thus, there is no way to compute distances between stimuli that do not appear together in a tested pair. In other words, the structure of the perceptual space is severely under-determined by the data. In future experiments such serial repetition of already-tested mixtures would be required to build up a data set to which the proposed method could be applied.

## Acknowledgements

We thank Krishnan Padmanabhan and Shreejoy Tripathy for helpful comments on the manuscript, and Dima Rinberg for thoughtful discussion.

## Additional information

### Author contributions

RCG, JBC, Conception and design, Analysis and interpretation of data, Drafting or revising the article

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

## Appendix 1

### Proof of sample size dependence

Here, we provide a more detailed statistical argument describing the framework's extreme sensitivity to incidental parameters. The crux of the statistical issue is this: the framework could only be valid if $d$, the estimated difference limen used in the calculation step, is a measure of olfactory resolution that converges to the true value of this quantity as more data is collected, that is, if it is *consistent*.

'Significantly discriminable' is a moving target dependent on sample size, choice of significance criterion, and correction for multiple comparisons. And $d$ is the only data-dependent value used in subsequent calculations (**Equation 1**), Together, this guarantees that the estimate of $z$ in (**Bushdid et al., 2014**) is a moving target as well, dependent on these same parameters. $d$ is generated by testing a number of null hypotheses, and is closely related to the fraction of these which are rejected. But the probability of and criteria for rejection of these null hypotheses depends critically on sample size and $\alpha$, the values that we explored in **Figure 3** and **Table 2**. Certainly, we would agree that there is nothing objectionable about the specific parameters chosen in (**Bushdid et al., 2014**). However, there is nothing objectionable about many other values for those parameters either.

In effect, calculating $d$ is somewhat like judging whether a coin meets a cutoff for being fair based on a series of tosses. It matters very much how many tosses one makes, and how much deviation from chance one is willing to tolerate before calling a coin unfair. If you have no particular reason to believe a coin is unfair, you might be disinclined to call it unfair if you observe $\frac{6}{10}$ (60%) heads, but probably not if you observed $\frac{600}{1000}$ heads (also 60%). However, if you own a casino, you might call 5100 heads in 10,000 (51%) evidence of an unfair coin. Whether the coin is fair is not something we directly measure, but rather we have more or less evidence for various degrees of fairness.

A similar situation applies in (**Bushdid et al., 2014**)'s analysis by considering its formal definition of $d$ (a definition we verified by reconstructing the critical figures from (**Bushdid et al., 2014**) in **Figure 2**. $d$ is defined as that inter-stimulus distance $D$ for which 50% of subjects can significantly discriminate a mixture class. By a mixture 'class' we denote the set of mixture pairs for which each mixture has the same number of total components ($N$) and each pair has the same number of distinct, non-overlapping components $D$ ($D = N - O$, see **Table 1**). For example, the mixture pair ($ABC$, $ABC$) would be a member of the class with $N = 3$ and $D = 1$ distinct components. We focus here on calculations pertaining to the number of tests $T$ per class, but the same argument is readily translated over to the number of subjects $S$.

To assess significant discriminability from chance, (**Bushdid et al., 2014**) used a two-tailed binomial test. Thus if a p-value is smaller than $\frac{\alpha}{2}$ then the subject is considered able to significantly discriminate from pairs in the mixture class. The p-value is given by 1 minus the cumulative binomial distribution function for $n = T$ trials, $k$ successes, and a probability of success equal to $\frac{1}{3}$, with $k$ corresponding to the number of subjects discriminating correctly, and $\frac{1}{3}$ to chance in a 3-way forced choice task. Thus, the subject's discrimination performance is significant if:

$$\alpha/2 > 1 - cdf_{binomial}\left(T, k, \frac{1}{3}\right) = \sum_{i=0}^{k} \binom{T}{i} \left(\frac{1}{3}\right)^i \left(\frac{2}{3}\right)^{T-i} \tag{3}$$

For $\alpha = 0.05$, $T = 20$ (as used in [**Bushdid et al., 2014**]), this inequality is satisfied for $k >= 11$. For each subject, $k$ might be any value between 0 and 20 depending on olfactory acuity. If $k >= 11$ for more than 50% of subjects, then the value of $D$ characterizing that mixture pair is necessarily $> d$. If $k >= 11$ for fewer than 50% of subjects, then $D < d$. If $k >= 11$ for exactly 50%

of subjects, then $D = d$. The actual estimate for $d$ is obtained by regression in the spirit of **Figure 2**.

What kind of subject can discriminate successfully 11 times out of 20? Consider a mixture class $X_{N,D}$ (characterized by $N$ and $D$), and a subject performance of $f_{N,D}$, corresponding to the proportion of mixtures correctly discriminated from a sample of size $T$. Note that $f_{N,D}$ is simply the abscissa of **Figure 1** from (**Bushdid et al., 2014**). A subject with $f_{N,D} = 0.55$ would get $k = T*f_{N,D} = 11$ out of $T = 20$ correct on average. So we can rewrite the inequality above as an equation:

$$1 - \alpha/2 = \sum_{i=0}^{f_{N,D}*T} \binom{T}{i} \left(\frac{1}{3}\right)^i \left(\frac{2}{3}\right)^{T-i} \tag{4}$$

If the above equation is satisfied, then the subject will be considered to be on the boundary between significantly discriminating and not significantly discriminating mixture pairs in the class. If half of subjects perform better than $f_{N,D}$, and half less, then half of subjects will be considered to significantly discriminate mixture pairs in the class (and half not), and so $d$ will be set equal to $D$. This is simply the definition of $d$.

The value $f_{N,D}$ for which that equation is satisfied depends upon $\alpha$ and $T$. $f_{N,D}$ is related to $N$ and $D$ through the data, and so the value of $D$ for which the equation is satisfied (i.e., $D = d$) depends upon $\alpha$, $T$, and the data. However, it is inappropriate for the discriminability limen to depend on $\alpha$ and $T$ in this way. As we showed above, this has serious consequences for the estimate of $d$, and therefore also for the estimate of $z$. It is what makes $z$ inconsistent.

**Figure 1—figure supplement 1** shows the relationship between the critical $f_{N,D}$, $T$, and $\alpha$. Note that this relationship is independent of the data. The data only determine how $f_{N,D}$ depends upon $D$ and consequently determines $z$. In summary, a smaller (larger) value of $\alpha$ or $T$ requires a much higher (lower) value of $f_{N,D}$ to satisfy the equation. This higher (lower) value of $f_{N,D}$ might only be found at a much larger (smaller) value of $D$, implying a much larger (smaller) value of $d$ and therefore a much smaller (larger) value of $z$.

With a sufficient number of subjects (or tests), even barely above chance performance can produce estimates of $z$ equal to the largest possible number of stimuli (**Figure 3** and **Figure 3—figure supplement 1**). In fact, this is guaranteed by **Equation 4**. The critical values of $f_{N,D}$ required for statistical significance will asymptotically approach $\frac{1}{3}$ (chance) as $T$ approaches infinity. The same principle applies to a consideration of changes to the number of subjects $S$, instead of the number of tests. This illustrates the core of the problem. Discriminating significantly above chance can be a very high bar or a very low bar depending on the parameters of the experiments and the analysis, including $S$, $T$, and $\alpha$.

## Can regularization solve this problem?

An alternative way to generate hypothetical data for larger values of $S$ or $T$ would be to imagine mean discrimination performance converging to the true population value as sample size increases. This has intuitive appeal, as surely the fraction discriminated should converge to $\frac{1}{3}$ for, say, identical stimuli, as the number of subjects approaches infinity. However, there is no clean way to generate hypothetical data in such a way for non-trivial cases, such as data where the mixtures are clearly discriminable, without knowing in advance what the population average is! If one could partition mixture classes into clearly discriminable and indiscriminable, and assume that the indiscriminable converge to $\frac{1}{3}$ discriminated, and the discriminable to same value larger than $\frac{1}{3}$, the resulting plot would show a clear limen boundary where the data departed from $\frac{1}{3}$. However it is likely that *all* mixture classes have at least some discriminable pairs, and even if that is only one pair out of a thousand, and only one subject out of a thousand can discriminate it, we would declare the class to be discriminable and the limen $d$ to be smaller than the $D$ for that class; we would likely do this for all classes, resulting in the conclusion (using **Equation 1**) that all stimuli are discriminable. This occurs because we need $d$ to be a property of the

stimulus space in general, not a quirk of a small fraction of mixtures. But this kind of *d* is elusive. So deriving *d* in this way won't justify the use of the subsequent sphere-packing framework.

## Correct bounds for the sphere packing problem

Here we elaborate on our claim that the 'trillions' figure is in fact an estimate for an *upper* bound, and not an estimate for a *lower* bound, as advertised in the title of (**Bushdid et al., 2014**), and throughout that report. The practical upshot of this is that even if one grants all other aspects of the framework, it still makes an unremarkable claim, one concerning how many olfactory stimuli humans can discriminate *at most*. Claiming to be *at least* 7 feet tall is a bold claim indeed. In contrast, claiming to be *at most* 7 feet tall is not a claim worth making. Providing a very high upper bound for the number of olfactory stimuli that humans can discriminate does nothing to advance our understanding of human olfactory ability. An expanded version of the following (including code for all calculations, and additional supporting plots) can be found at http://github.com/rgerkin/trillion.

The minimum possible number of discriminable stimuli occurs when olfactory resolution is as bad as possible. This occurs when *d*, the discriminability limen, is equal to *N*, the number of components. This means that even when every component is substituted in a mixture, discrimination is still hard or impossible. In the visual system, the equivalent situation would be a discriminability limen that spans the entire range of visible wavelengths. Here, the 'sphere packing' calculation for such a limen produces the sensible result that there is only one resolvable color percept, only if one large 'sphere' of diameter *d* spans these wavelengths.

Curiously though, using **Equation 1** (implemented exactly from [**Bushdid et al., 2014**]), and setting *d* = *N*, we obtain the value 4561 (for *N* = 30), that is, we would estimate that there are thousands of discriminable stimuli. This can be confirmed in (**Bushdid et al., 2014**) by inspecting the y-intercept of its **Figure 4C,D**, or in our **Figure 6A**. Clearly this is problematic since *d* = *N* should correspond to worst possible performance.

Perhaps this simply traces back to ambiguity in how the limen *d* is defined, and how end-points are treated. For example, we could adopt the convention that a limen of 20 nm in color vision means that stimuli separated by 20 nm are *just discriminable*; alternatively, that such stimuli are the farthest that are *still indiscriminable*, which is slightly different. Along similar lines, one interpretation of a limen of *d* = *N* is that stimuli are only discriminable when *all components* are replaced. If this is the case, then the estimate should be equivalent to the number of ways replacing all *N* components. For a library of size *C*, this can be done in $\frac{C}{N} = \frac{128}{30} \sim 4$ ways, which is still clearly discordant with the result from **Equation 1** (4561). While 4561 is not remarkably high, it is clearly inconsistent with common sense. Furthermore, the edge-case, *d* = *N* behavior of **Equation 1** as *C* increases produces increasingly implausible results, which are independent of the data: they are guaranteed by the equation (see **Figure 5**).

In an effort to understand the behavior of **Equation 1**, which is advertised in (**Bushdid et al., 2014**) to provide the number of discriminable stimuli, one can ask 'what value *d* > *N* is needed to produce the result that there is only 1 discriminable stimulus?' (that is, that only 1 'ball' occupies the range of discriminable stimuli). The answer is that *d* must be equal to 2*N* before **Equation 1** will provide the result of 1 discriminable stimulus. But clearly *d* = 2*N* is impossible since *d* <= *N* by definition. This indicates that there is potentially a spurious factor of two somewhere in the calculation for the lower bound.

## Correcting the lower bound

A common-sense interpretation of a scenario where *no* N-component mixtures differing in all components (*d* = *N*, the largest possible limen) can be discriminated from one another is that there is only one resolvable percept (not thousands). This interpretation will be secured using the following corrected equation for the lower bound:

$$z(d) = \frac{\binom{C}{N}}{ball(d)} \qquad (5)$$

instead of the equation used in (**Bushdid et al., 2014**) and in the main text here as **Equation 1**:

$$z(d) = \frac{\binom{C}{N}}{ball(d/2)} \qquad (6)$$

**Figure 6** shows the behavior of the estimated number of discriminable stimuli, $z$, as a function of the discriminability limen, $d$, for these two equations. Note three important features in the behavior of **Equation 5** (as seen in **Figure 6B**):

- First, for worst possible discrimination ($d = N$) the minimal number of discriminable stimuli is 1, which is sensible.
- Second, the maximal number of discrimianble stimuli is still equal to the total number of mixtures that can be constructed from the library (as with the original **Equation 6**), which is again sensible.
- Third, and most importantly, the number of discriminable stimuli estimated from the data is now orders of magnitude smaller, and within the folk wisdom range.

Note that **Equation 5** will still underestimate $z$, given $d$ and the acceptance of the remainder of the assumptions in the framework. After all, it is a lower bound. But similarly, **Equation 6** will always overestimate it. This is well-known from the theory of error-correcting codes, where **Equations 5, 6** represent lower and upper bounds on the solution to a homologous problem in coding theory (**MacWilliams and Sloane, 1977**). These bounds are essentially the constant-weight versions of the Gilbert-Varshamov and Hamming bounds, respectively, and have been proven mathematically; specifically, the lower bound is due to Levenshtein (**Levenshtein, 1971**; **Jiang and Vardy, 2004**), and the upper bound is slightly weaker version of that developed by Freiman, Berger, and Johnson (**Freiman, 1964**; **Agrell et al., 2000**).

