## [Decision Letter]

Thank you for sending your work entitled “The number of olfactory stimuli that humans can discriminate is still unknown” for consideration at *eLife*. Your article has been favorably evaluated by Eve Marder (Senior editor) and two reviewers, one of who, Alexander Borst, is a member of our Board of Reviewing Editors.

The Reviewing editor and the other reviewer discussed their comments before we reached this decision. Based on the two reviews below and our discussions, we will be pleased to accept your work for publication.

Reviewer #1:

This is a timely study shedding light from an interesting statistical angle on the paper by Bushdid et al., published in 2014. As the authors demonstrate, the formula used in the previous account poses an upper bound, not a lower bound, on the number of discriminable odorants. Furthermore, the formula used depends in a very sensitive way on various factors like the number of participants and the number of test odors used. In conclusion, the statement from the previous study is untenable.

Reviewer #2:

This manuscript is a critique of the methods used in a recent paper by Bushdid et al. in which the authors purport that humans can discriminate between at least a trillion odors. Gherkin and Castro here dissect methodically the reasons why this conclusion is unwarranted and provide a limpid assessment of the mistakes made in the original study. (The critique is carefully limited to methodological issues, and makes no statement as to whether the initial claim might eventually be shown to be true or not.)

This paper is a treasure of clarity and thoroughness. The points made in the main manuscript are further expanded in an appendix, and the suggestion to improve the estimate via constructing a “continuous” perceptual map of odors is excellent.

[Editors’ note: minor issues and corrections have not been included, so there is not an accompanying Author response.]